# Mouse Models of Achromatopsia in Addressing Temporal “Point of No Return” in Gene-Therapy

**DOI:** 10.3390/ijms22158069

**Published:** 2021-07-28

**Authors:** Nan-Kai Wang, Pei-Kang Liu, Yang Kong, Sarah R. Levi, Wan-Chun Huang, Chun-Wei Hsu, Hung-Hsi Wang, Nelson Chen, Yun-Ju Tseng, Peter M. J. Quinn, Ming-Hong Tai, Chyuan-Sheng Lin, Stephen H. Tsang

**Affiliations:** 1Department of Ophthalmology, Edward S. Harkness Eye Institute, Columbia University Irving Medical Center, New York, NY 10032, USA; wang.nankai@gmail.com (N.-K.W.); aleckliu418@gmail.com (P.-K.L.); yk2837@cumc.columbia.edu (Y.K.); Slevi@wesleyan.edu (S.R.L.); huangyiwen780122@gmail.com (W.-C.H.); cwheye2118@gmail.com (C.-W.H.); ethanwang1117@gmail.com (H.-H.W.); nelson720720@gmail.com (N.C.); pq2138@cumc.columbia.edu (P.M.J.Q.); 2Department of Ophthalmology, Kaohsiung Medical University Hospital, Kaohsiung Medical University, Kaohsiung 80756, Taiwan; 3School of Medicine, College of Medicine, Kaohsiung Medical University, Kaohsiung 80708, Taiwan; 4Institute of Biomedical Sciences, National Sun Yat-sen University, Kaohsiung 80424, Taiwan; 5Department of Pathology & Cell Biology, Columbia University Irving Medical Center, New York, NY 10032, USA; yt2566@cumc.columbia.edu (Y.-J.T.); csl5@cumc.columbia.edu (C.-S.L.); 6Herbert Irving Comprehensive Cancer Center, Columbia University Irving Medical Center, New York, NY 10032, USA; 7Center for Neuroscience, National Sun Yat-sen University, Kaohsiung 80424, Taiwan; 8Graduate Program in Marine Biotechnology, National Sun Yat-sen University, Kaohsiung 80424, Taiwan; 9Jonas Children’s Vision Care, and Bernard and Shirlee Brown Glaucoma Laboratory, Columbia Stem Cell Initiative, Departments of Ophthalmology, Pathology and Cell Biology, Institute of Human Nutrition, Vagelos College of Physicians and Surgeons, Columbia University, New York, NY 10032, USA

**Keywords:** achromatopsia, CNGA3, CRISPR

## Abstract

Achromatopsia is characterized by amblyopia, photophobia, nystagmus, and color blindness. Previous animal models of achromatopsia have shown promising results using gene augmentation to restore cone function. However, the optimal therapeutic window to elicit recovery remains unknown. Here, we attempted two rounds of gene augmentation to generate recoverable mouse models of achromatopsia including a *Cnga3* model with a knock-in stop cassette in intron 5 using *Easi*-CRISPR (*E*fficient *a*dditions with *s*sDNA *i*nserts-CRISPR) and targeted embryonic stem (ES) cells. This model demonstrated that only 20% of CNGA3 levels in homozygotes derived from target ES cells remained, as compared to normal CNGA3 levels. Despite the low percentage of remaining protein, the knock-in mouse model continued to generate normal cone phototransduction. Our results showed that a small amount of normal CNGA3 protein is sufficient to form “functional” CNG channels and achieve physiological demand for proper cone phototransduction. Thus, it can be concluded that mutating the *Cnga3* locus to disrupt the functional tetrameric CNG channels may ultimately require more potent STOP cassettes to generate a reversible achromatopsia mouse model. Our data also possess implications for future *CNGA3*-associated achromatopsia clinical trials, whereby restoration of only 20% functional CNGA3 protein may be sufficient to form functional CNG channels and thus rescue cone response.

## 1. Introduction

Achromatopsia, also known as rod monochromatism, is a rare autosomal recessive cone dysfunction disorder that manifests at birth or during early infancy. With an estimated incidence of 1 in 30,000, affected individuals typically present with amblyopia (or lazy eye), nystagmus, marked photophobia, decreased visual acuity, hyperopic refractive error, and very poor or no color vision [1,2]. Characteristic electrophysiologic findings include no detectable cone function but a normal or near-normal rod response [1,2,3]. Several gene mutations have been reported in achromatopsia, which include the genes that encode components of the cone phototransduction cascade. These genes include *CNGA3*, *CNGB3*, *GNAT2*, *PDE6C*, and *PDE6H* [4,5,6,7], as well as the *ATF6* gene that encodes ATF6 protein in response to endoplasmic reticulum stress [8]. *CNGA3* and *CNGB3* mutations are the two most common causes and are responsible for approximately 80% of all achromatopsia cases. The prevalence of mutations in *CNGA3* range from 25–28% in Western populations to 60% in Middle Eastern and Arabic populations [1,2,6,9,10,11].

Photoreceptor cyclic nucleotide-gated (CNG) channels, localized on the plasma membrane of photoreceptor outer segments, are strictly ligand-gated and regulated by cGMP. The CNG channels are the last component in the activation phase of phototransduction [12,13]. As part of this process, the CNG channel stays open in the dark, permitting the transport of cations. As the light enters the eye, it induces the blockage of these channels, inhibiting the inward flow of cations, thereby activating the phototransduction pathway.

In mammals, the tetrameric CNG channels are composed of structurally similar A and B subunits. The rod channel consists of CNGA1 and CNGB1 subunits, whereas the cone channel is composed of CNGA3 and CNGB3 subunits. The most common channel mutation is a missense mutation in the *CNGA3* gene, affecting only a single amino acid residue of the protein [9,10]. These mutations then impair the channel’s ability for hyperpolarization within the cone phototransduction cascade [14]. While both subunits form the structural components of the CNG channel, it is A subunits that are believed to confer the principal channel properties, whereas the B subunits are essential for proper outer segment localization and contribute specific biophysical properties to the native channel complex [15,16]. As a result, the aforementioned missense mutations in the *CNGA3* gene cause the main channel impairments.

Previous studies have suggested that gene therapy can effectively restore cone function in CNGA3-deficient mouse models [17,18]. Recently, phase I/II retinal gene therapy trials for *CNGA3*-mediated achromatopsia demonstrated some level of cone photoreceptor function restoration in adult patients [19,20]. These experiments are timely in light of four ongoing adeno-associated virus (AAV) gene supplementation trials in adult cone dystrophy patients; in one of these trials, early electroretinogram (ERG) efficacy data suggested that cone function was not improved by AAV::*CNGA3* gene augmentation [19,20,21]. However, it is still unknown whether treating younger patients will result in greater functional gains by treating patients in a timely manner during the optimal therapeutic window for amblyopia, a visual impairment caused by the inability of the eyes and brain to work together with no causative structural abnormalities in the eyes or visual pathways. Amblyopia occurs early in life when the developing visual cerebral cortex is unable to receive clear images formed on the retina through the lateral geniculate and visual pathways [22]. Since the plasticity of the visual cortex development decrease as age increases, the amblyopia may be no longer reversible after a certain point [23]. As such, it may be critical to treat patients with achromatopsia prior to the critical window, in order to avoid the potential limiting factor of amblyopia. Therefore, we attempted to design an innovative tool to model gene therapy: A *Cnga3^floxed-STOP^* allele that can be conditionally reverted from mutant to wild-type (WT) upon tamoxifen activation of a novel Cre recombinase driver (*Arr3^CreERT2^*). Herein, we present the findings from our studies, applying this *Cnga3^floxed-STOP^* allele tool to study factors limiting outcomes in gene therapy, including but not limited to temporal factors that determine which patients are treatable.

## 2. Results

### 2.1. Functional Electroretinography of Cnga3^cpfl5^ Mice

Before identifying which genes would be used to create our potentially reversible achromatopsia mouse model, we tested ERG in B6.RHJ-*Cnga3^cpfl5^*/BocJ mice (JAX Stock No. 005978, hereafter *Cnga3^cpfl5^* mice) [24] at one-month-old to validate whether dysfunction of the Cnga3 protein in mice could phenocopy the ERG features exhibited in patients with achromatopsia. Specifically, these *Cnga3^cpfl5^* mice carry a spontaneous A to G substitution in exon 5 and, at 5.5 months of age, show no *b*-wave response in light-adapted ERG and decreased rod-mediated ERG responses [24]. Additionally, our results in scotopic serial intensity ERG also showed statistically significant decreased *b*-wave amplitudes in *Cnga3^cpfl5^* mice, as compared to age-matched C57BL/6J mice (JAX Stock No. 000664) (Figure 1, left). Photopic serial intensity ERG in *Cnga3^cpfl5^* mice showed nearly extinguished *b*-wave amplitudes, a statistically significant decrease as compared to the *b*-wave amplitudes in age-matched C57BL/6J mice (Figure 1, right). Therefore, we chose the *Cnga3* gene as our target to generate our potentially reversible achromatopsia mouse model. 

### 2.2. Generation of Cnga3^floxed-miniSTOP^ Mice Using Easi-CRISPR

To generate a novel, reversible mouse model of achromatopsia, we used *Easi*-CRISPR (Efficient additions with ssDNA inserts-CRISPR) technology [25] to knock-in *floxed-miniSTOP* cassettes (393 bps) in intron 5 of the *Cnga3* gene (Figure 2a). Three target sequences of sgRNA (Table 1) were designed using Benchling [26], and subsequently tested using Cas9 protein/sgRNA ribonucleoprotein complexes (Cas9-RNPs) for their cutting efficiency while forming the ribonucleoprotein by in vitro digestion (IVD) assay (Figure 2b, left). sgRNA#3 showed the best cutting efficiency in the IVD test and was selected for further pronuclear injection with Cas9 protein and single-stranded oligodeoxynucleotides (ssODN). The ssODN (653 bases) includes 393 bases of *floxed-miniSTOP* cassette (393 bps) and homologous sequences at both ends (115 bases and 145 bases) corresponding to the cutting site of sgRNA#3 (Figure 2b, right). The floxed-miniSTOP cassette, which is flanked by loxP, contains two splicing acceptors, a three-phase translation stop, and one polyadenylation signal.

Twelve pups were generated successfully after pronuclear injection of ssODN, the Cas9 protein, and sgRNA#3. We used the polymerase chain reaction (PCR) with primers (Table 2) to screen the 5′ and 3′ junction of the miniSTOP and identified 7 mice (#2, 4, 5, 6, 9, 10, 11) of the 12 pups carrying the knock-in floxed-miniSTOP cassettes (Figure 2c, left). We further used primers 5′F and 3′R to screen for WT and knock-in floxed-miniSTOP cassette alleles in these 12 pups and found that only the knock-in floxed-miniSTOP cassettes allele could be amplified from founder #6 (Figure 2c, left). The photopic serial intensity ERG showed decreased *b*-wave amplitudes in the #6 founder mouse (homozygous-like knock-in) compared to the #2 (heterozygous-like knock-in) and #3 (no knock-in) founder mice (Appendix A). This male founder (#6) mouse was outcrossed with a WT C57BL/6J mouse to produce the first filial generation hybrid offspring (F1).

PCR screening of the 5′ and 3′ junction of the five F1 mice revealed that four F1 mice carried the knock-in floxed-miniSTOP cassette allele. Among these four F1 mice (#1, 2, 4, 5), the PCR amplification with primers 5′F and 3′R could amplify both knock-in floxed-miniSTOP cassette and WT alleles. The F1 animals (#1 and #5) were then intercrossed to produce homozygous F2 offspring to confirm penetrance. PCR screening using primers 5′F and 3′R on six F2 mice showed two wild-type, two heterozygous, and two homozygous of *Cnga3^floxed-miniSTOP^* mice (hereafter *Cnga3^floxed-miniSTOP^* mice). (Figure 2d, right). The direct sequence from #3 F2 mice successfully showed the knock-in floxed-miniSTOP cassette in the intron 5 of the *Cnga3* gene (Figure 2d, bottom).

### 2.3. Functional Electroretinography and Histology of Cnga3^floxed-miniSTOP^ Mice

The scotopic and photopic serial intensity ERG of *Cnga3^floxed-miniSTOP^* mice revealed no decrease in *b*-wave amplitudes compared to the *Cnga3^+/+^ and Cnga3^floxed-miniSTOP/+^* mice from the same litter at 1-month old (Figure 3, left). To further validate whether the *Cnga3^floxed-miniSTOP^* mice could be a slow degeneration in cone response, we tested the scotopic and photopic serial intensity ERG at 3 months old, which showed no statistically significant difference in *b*-wave amplitudes when comparing *Cnga3^floxed-miniSTOP^* and litter-control *Cnga3^+/+^* mice (Figure 3, right). Histology also showed no difference in the thickness of the outer nuclear layer between *Cnga3^+/+^, Cnga3^floxed-miniSTOP/+^*, and *Cnga3^floxed-miniSTOP^* mice from the same litter at 6 months old (Appendix A).

### 2.4. Functional Electroretinography of Cnga3^floxed-STOP^ Mice

Since the cone responses in *Cnga3^floxed-STOP^* mice remained robust, we moved forward with a second attempt to generate another reversible mouse model of achromatopsia by inserting a “*floxed*-*STOP2886*” (hereafter “*floxed*-*STOP*”) cassette in the intron between exon 5 and exon 6 of the *Cnga3* gene using embryonic stem (ES) cell gene targeting (Figure 4). After confirming germline transmission, we intercrossed *Cnga3^floxed-STOP/+^* mice to generate *Cnga3^+/+^*, *Cnga3^floxed-STOP/+^*, and *Cnga3^floxed-STOP^* mice in the same litter. For each of these mouse models, we then performed ERG testing to identify any changes in the scotopic and photopic responses.

Overall, scotopic and photopic serial intensity ERG of *Cnga3^floxed-STOP^* mice showed no decrease in *b*-wave amplitudes compared to the *Cnga3^+/+^ and Cnga3^floxed-STOP/+^* mice from the same litter at 1-month old (Figure 5, left). To further validate whether the *Cnga3^floxed-STOP^* mouse model experiences a slow degeneration in cone response, we tested scotopic and photopic serial intensity ERG at 6 months old. Although the photopic *b*-wave amplitudes are smaller in *Cnga3^floxed-STOP^* mice, the difference is not statistically significant (Figure 5, right).

### 2.5. Quantitative Real-Time RT-PCR of Cnga3^floxed-STOP^ Mice

Due to the prokaryotic transcriptional termination in the “*floxed*-*STOP*” cassette, we investigated whether the “*floxed*-*STOP*” cassette could efficiently terminate transcription of the *Cnga3* gene. We designed primers (Cnga3 3′) that span intron 5 and anneal to sequences in exons 5 and 6 (Figure 6A). We found that the *Cnga3* mRNA was decreased by 80% in *Cnga3^floxed-STOP^* mice compared to litter-control *Cnga3^+/+^* mice.

To further examine whether there is an upregulation of *Cnga3* transcript upstream of the floxed-STOP cassette in compensation for the decreased transcription of *Cnga3* downstream of the floxed STOP cassette, we designed primers (Cnga3 5′) that span intron 2 and anneal to sequences in exons 2 and 3 (Figure 6A). Despite an increasing trend, the level of *Cnga3* transcription product upstream of the floxed-STOP cassette showed no statistically significant difference among the three groups.

In *Cnga3* and *Cngb3* knock-out mice, there are reports of decreased CNGB3 and CNGA3 proteins, respectively [27,28]. Next, we examined whether decreased *Cnga3* mRNA in our *Cnga3^floxed-STOP^* mice upregulates *Cngb3* transcription as compensation. The results showed no significant difference in the *Cngb3* transcript in our *Cnga3^floxed-STOP^* mice compared to litter-control *Cnga3^+/+^* mice (Figure 6A). Primers used in the transcriptome analysis are listed in Table 3.

### 2.6. Immunoblotting of CNGA3 in Cnga3^floxed-STOP^ Mice

Because there was 20% mRNA expression in *Cnga3^floxed-STOP^* mice, we further examined the CNGA3 protein expression by Western blotting. We used an anti-CNGA3 antibody that targeted the C-terminal of the CNGA3 protein (amino acid residues 482–605) translated from exon 6 after the floxed-*STOP* cassette (Figure 6B). There was reduced CNGA3 protein expression in *Cnga3^floxed-STOP^* mice compared to litter-control wild-type mice, while there was no significant decrease in CNGA3 protein expression in *Cnga3^cpfl5^* mice. As expected, the reduced CNGA3 protein in *Cnga3^floxed-STOP^* mice was in line with the reduced *Cnga3* mRNA transcription, and the missense mutation in *Cnga3^cpfl5^* mice did not affect the CNGA3 protein expression (Figure 6B). Because it has been reported that the reduction of cone response approximated the reduction of cone proteins rather than cone numbers [29], we examined the ARR3 protein expression by Western blotting. We found there was no significant change in ARR3 protein expression in our *Cnga3^floxed-STOP^* mice (Figure 6B), since there was no reduction of cone response in our *Cnga3^floxed-STOP^* mice. However, there was no significant ARR3 protein reduction in *Cnga3^cpfl5^* mice, which have extinguished cone responses (Figure 6B). Therefore, the reduction of cone response was not related to cone ARR3 protein expression in *Cnga3^cpfl5^* mice.

Although we found decreased CNGA3 protein expression in our *Cnga3^floxed-STOP^* mice, there is no decrease in cone ERG responses. To investigate whether the compensatory CNGB3 protein upregulation could be a possibility in forming homoretrameric CNG channels that maintain normal cone ERG response, we examined the CNGB3 protein expression *Cnga3^floxed-STOP^* mice, which showed no obvious change of CNGB3 protein expression compared to litter-control wild-type mice (Figure 6C).

### 2.7. Functional Electroretinography of Compound Heterozygous Cnga3^cpfl5/floxed-STOP^ Mice

Given that the “*STOP cassette*” can knock down ~80% *Cnga3* mRNA transcription in *Cnga3^floxed-STOP^* mice, we subsequently asked whether compound heterozygous mice could be an alternative recoverable mouse model that could present early cone dysfunction as seen in *Cnga3^cpfl5^* mice. Thus, we crossed *Cnga3^floxed-STOP^* with *Cnga3^cpfl5^* mice to generate *Cnga3^cpfl5^/Cnga3^floxed-STOP^* (hereafter *Cnga3^cpfl5/floxed-STOP^* mice) compound heterozygous mice and subsequently tested the functional ERG in these compound heterozygous models. The resulting scotopic and photopic serial intensity ERGs from these compound heterozygous mice showed relatively normal ERG tracings compared to C57BL/6J mice at the same age (postnatal day 31, P31) (Figure 7).

## 3. Discussion

### 3.1. Easi-CRISPR

CRISPR/Cas9 technology has become more and more popular for gene editing in mice. Compared to embryonic stem cells targeting, *Easi*-CRISPR shortens the time to achieve the genetic modulated mice. Although there is still a limitation in the size of knock-in gene fragments, it could edit two alleles of the gene at the same time, whereas traditional ES cell targeting could only target one allele. In our study, 7 of the 12 founder mice had a successfully generated knock-in of the 393-base-floxed-miniSTOP cassette. We initially thought that the founder #6 mouse was a homozygous knock-in floxed-miniSTOP mouse based on its genotyping results (Figure 2C). However, when we crossed this founder to a WT C57BL/6J mouse, we were unable to amplify the knock-in allele in one (#3 mouse) of the five F1 mice (Figure 2). This indicates that the founder #6 mouse carries one allele of knock-in floxed-miniSTOP and the other allele is a deletion that extends beyond the region of 5′F and 3′R primers. The major concern of CRISPR/Cas9 is that DNA breaks introduced by the single-guide RNA/Cas9 frequently result in deletions, which could extend over many kilobases [30]. While this concern may limit its application in gene therapy directly in vivo, this potential issue from *Easi*-CRISPR could be solved by breeding the founder mice with WT mice to generate offspring with the correctly edited genes.

### 3.2. ERG Setting for Testing Mouse Models

The international society for clinical electrophysiology of vision (ISCEV) has established standard electroretinography protocols [31] for humans in terms of the period for dark and light adaptation as well as specific flash intensities, which not only can detect rod, cone, and combined rod and cone responses, but also allow the results to be interpretable and comparable using different machines from different centers. When describing a mouse model with cone cell dysfunction, it is essential to use a photopic ERG machine capable of detecting isolated cone response/function under conditions that saturate the rod system [32,33,34]. To eliminate the residual rod response, light stimulus under background illumination after 10 min of light adaptation would be preferred [32,33,34]. Several genetic modified and spontaneous mutant mice have been reported with abnormal cone function. However, the settings of ERG from these studies are different, which limit its comparison between these mouse models. In our study, we applied the same protocols (serial intensities after 10 min of light adaptation) on our genetically modified mice and *Cnga3^cpfl5^* mice, from which we confirmed the absent cone responses in *Cnga3^cpfl5^* mice. In the future, there is a need for standardized ERG protocols for characterizing electrophysiologic function in mice.

### 3.3. Splicing Acceptors and Score Prediction

The “*floxed-STOP*” cassette used in our study had been tested in another project (unpublished), from which RT-PCR showed absent RNA transcription downstream of the “*floxed-STOP*” cassette. However, in our study, we could still detect ~20% RNA transcription spanning the “*floxed-STOP*” cassette. To further predict the efficiency of splicing acceptors in the location of the intron, we applied two prediction algorithms for the analysis of the splicing effect (Table 4). We found the splicing acceptor scores of exon 6 are 0.97 and 0.99 using two different prediction algorithms. After inserting the “*floxed-STOP*” there is no change in the splicing scores of each exon of the *Cnga3* gene. The splicing acceptor scores, using two prediction algorithms of the “*floxed-STOP*” cassette, are 0.97 and 1 (Table 4). Although the splicing acceptor scores inside the “*floxed-STOP*” cassette are competitive (0.97 and 1), some spliceosomes may skip the splicing acceptors inside the “*floxed-STOP*” cassette and jump to the robust splicing acceptor of the last exon of the *Cnga3* gene (0.97 and 0.99).

### 3.4. Ion-Conducting Subunit of the Channel

In all six mammalian CNG channel subunits, only CNGA1, CNGA2, and CNGA3 can form functional homotetrameric channels in heterologous expression systems. The remaining CNGA4, CNGB1, and CNGB3 subunits do not assemble to functional homotetramers [15,16,35]. In cone cells, the CNG channels are heterotetramers assembled by either 3A:1B or an equal 2A:2B [36,37,38,39]. Although CNGB3 shares a common topology with CNGA3, and is also composed of a pore-forming region, CNGB3 does not form a homotetrameric functional channel, in contrast to CNGA3 [40,41]. Notably, in *Cnga3^−/−^* mice, there is missing CNGB3 protein expression in the cone outer segment at 3 months old [28], whereas in *Cngb3^−/−^* mice, there were undetectable CNGA3 protein and dramatically reduced *Cnga3* mRNA [27]. Therefore, the CNGA3 and CNGB3 proteins are closely dependent on each other to be normally targeted in cone outer segments [27,28]. In our study, we noticed a decrease in CNGA3 protein expression and mRNA in the *Cnga3^floxed-STOP^* retina. However, there is no significant change in the CNGB3 protein and mRNA levels. Therefore, the normal cone response ERG in our *Cnga3^floxed-STOP^* mice was not caused by increased CNGB3 protein compensation, which may indicate that the 20% of the normal CNGA3 protein could be sufficient for normal phototransduction in cone cells.

### 3.5. ERG Cone Response Corresponds to CNG Channel Function?

It remains unclear whether the ERG cone response corresponds to the numbers of alive cone cells or proteins in these cone degeneration mouse models. In *Cnga3^−/−^* mice, there is no cone response at 2 months old; however, more cone cells die at 12 months old [28]. In *Cngb3^−/−^* mice, the residual cone response was not comparable to the percentage of remaining healthy cone cells because the cone response was only approximately 30% of the WT level, when approximately 80% of cones remained at 1 month old [29]. Interestingly, this study also found that the reduction of cone response (approximately 70–80%) did approximate the reduction of cone proteins such as M-OPSIN, GNAT2, and ARR3 [29]. Both *Cnga3^cpfl5^* and *Cngb3^cpfl10^* mice are spontaneously mutant mice and have been reported to show an absence of a cone response at early-onset and, gradually, cone cell death [14,24]. In our study, we found no reduction of the ARR3 protein in 1-month-old *Cnga3^cpfl5^* mice (Figure 6) when the cone responses are already extinguished. The missense mutations in both *Cnga3^cpfl5^* and *Cngb3^cpfl10^* mice result in a dysfunctional CNG membrane channel, which subsequently impairs hyperpolarization in the cone phototransduction cascade at the early stage, prior to cone cell death. The absence of a cone response in *Cnga3^−/−^* mice is due to the complete absence of CNGA3 protein to form a functional CNG channel, whereas the CNGB3 protein cannot form a channel themselves. The residual cone response in *Cngb3^−/−^* mice is due to CNGA3 even though decreased expression can form homotetrameric channels by themselves. Taken together, we believe the cone the ERG response corresponds to the normal cone phototransduction, which requires “functional” CNG channels from normal membrane proteins, rather than residual cone cells or “levels” of cone proteins.

## 4. Materials and Methods

### 4.1. Mouse Care and Housing

All mice used in this study were handled in accordance with the Statement for the Use of Animals in Ophthalmic and Vision Research of the Association for Research in Vision and Ophthalmology, and all experiments were approved by the Institutional Animal Care and Use Committee of Columbia University Irving Medical Center. C57BL/6J (JAX Stock No. 000664, hereafter *B6J* mice) and B6.RHJ-*Cnga3^cpfl5^*/BocJ mice (JAX Stock No. 005978, hereafter *Cnga3^cpfl5^* mice) were purchased from the Jackson Laboratory (Bar Harbor, ME, USA) and were housed at a local animal facility under a 12 h light/12 h dark cycle. All mice used in this study were in the *B6J* background and had been genotyped to confirm the absence of the *rd8* and *rd1* mutations because these mutations are present in vendor lines and confound ocular-induced mutant phenotypes [42,43].

### 4.2. Generation of Cnga3^floxed-miniSTOP^ Mice Using Easi-CRISPR

#### 4.2.1. sgRNAs

The guide sequence was designed using the online tool: Benchling [26]. Three sgRNA sequences were selected with lower off-target scores and higher on-target scores. The sequence of these three sgRNAs used in this study are shown in Table 1.

#### 4.2.2. In Vitro Digestion (IVD) Protocols

To test the cutting efficiency of sgRNA while forming the ribonucleoprotein with the Cas9 protein (NEB #M0386), we first amplified a DNA substrate (762 bps) from genomic DNA of B6J mouse using the primers: 5′-AAACCAAGGCCCCGAGTCCATATTC-3′ (forward) and 5′-TGCCCTTCAGGATCCATAGAGCAAC-′ (reverse). Then, we used the IVD protocol from the NEB protocol (https://international.neb.com/protocols/2014/05/01/in-vitro-digestion-of-dna-with-cas9-nuclease-s-pyogenes-m0386; accessed on 8 August 2019 ). We chose the sgRNA with the best cutting efficiency from IVD results for generating knock-in mice using *Easi*-CRISPR technology.

#### 4.2.3. Single-Stranded Oligo Donor (ssODN) Generation:

We amplified two PCR products: 1811 bp upstream (fragment A) and 2001 bps downstream (fragment C) of the cutting site from the selected sgRNA (#3). Fragment B containing *floxed-miniSTOP* (393 bps) was amplified from a plasmid that was synthesized by GenScript. The *floxed-miniSTOP* sequence is shown in Appendix A. Fragments A, B, and C were combined using the Gibson Assembly Cloning Kit (#E510, NEB), and then ssODN was generated using Guide-it™ Long ssDNA Production System (#632644, TaKaRa Bio USA, Mountain View, CA, USA). Three hundred and ninety-three bases of the *floxed*-*miniSTOP* cassette and homologous sequences at both ends (115 bases and 145 bases) corresponded to the cutting site of sgRNA#3 (Figure 2b, right). The ss-ODN sequence is shown in Appendix A. *Easi*-CRISPR was performed using the conventional pronuclear injection by Dr. Chyuan-Shen Lin, as previously described [44].

### 4.3. Generation of Cnga3^floxed-STOP^ Mice Using ES Cell Gene Targeting

We inserted a “*floxed*-*STOP2886*” (hereafter “*floxed*-*STOP*”) cassette in the intron between exon 5 and exon 6 of the *Cnga3* gene in the bacterial artificial chromosome (BAC) clone (RP23-149L22) using BAC recombineering (Figure 4). The floxed*-STOP2886* cassette contains four splicing acceptors, neomycin resistance, three polyadenylation signals, and the prokaryotic transcriptional termination that is designed to prevent gene expression at the level of transcription. The “*floxed*-*STOP*” sequence is shown in Appendix A. A gene-targeting vector was constructed by retrieving 1919 bps upstream of and 3474 bps downstream of the *floxed-STOP* cassette insertion. The gene-targeted vector was linearized with Asc1 and electroporated into KV1 (129B6N hybrid) ES cells. After G418 selection, six targeted ES cell clones were identified. One of the targeted ES cells was then injected into C57BL/8J blastocysts to generate chimeric mice. The subsequent male chimeras were crossed with C57BL/6J females for germline transmission.

### 4.4. Genotype

Mice were genotyped and verified using PCR analysis. Tail genomic DNA was isolated and amplified by PCR, using the primer set targeting the junction of the knock-in fragment (Table 2). Reactions contained 200 ng of genomic DNA, 0.25 μM primers, 200 μM dNTPs, 1.0 U of *Taq* polymerase, and 1 × PCR buffer (NEB ThermoPol, Ipswich, MA, USA) in a volume of 20 μL. The PCR products were resolved via 0.8% agarose gel electrophoresis using Gel Red (Invitrogen/Life Technologies, Waltham, MA, USA) as the visualizing dye. The DNA bands were visualized using an iBright Imaging System (Invitrogen).

### 4.5. Electroretinography

Before beginning the ERG procedure, mice were dark adapted for >12 h, and all subsequent scotopic ERG procedures were performed under dim red illumination, as previously described [45]. The mouse was anesthetized via intraperitoneal injection with 0.1 mL of a mixed solution (1 mL of ketamine at 100 mg/mL and 0.1 mL of xylazine at 20 mg/mL in 8.9 mL of PBS) per 10 g of body weight, and the pupils were dilated with topical 2.5% phenylephrine hydrochloride and 1% tropicamide. As part of the setup, the mice were positioned on a heated mat in front of the testing console (Colordome; Diagnosys LLC; Lowell, MA, USA), which also generated and controlled the light stimulus. The test protocol consisted of eight dark-adapted and eight light-adapted steps. The light intensities of the stimuli used for scotopic serial intensity ERG were –6.0, –5.0, –4.0, –3.0, –2.0, –1.0, 0.0, and 1.0 log cd·s/m^2^ in sequence. The intervals between each stimulus varied from 2 to 30 s, and the number of repeats varied from 20 to 3 times, for their respective light intensity. After the completion of dark-adapted recordings, the animals were exposed to a full-field 30 cd/m^2^ white background for 10 min; subsequent steps were delivered on top of this continuous background. The single-flash stimuli applied after light adaptation consisted of –1.0, 0.0, 0.3, 0.6, 1.0, 1.2, 1.48, 2.0, and 2.3 log cd·s/m^2^. The intervals between each stimulus varied from 1 to 10 s, and the number of repeats varied from 10 to 3 times. A digital band-pass filter ranging from 0.3 to 300 Hz was used to isolate signals after the waves were recorded. The a-wave amplitude was measured from the baseline to the trough of the *a*-wave, and the *b*-wave amplitude was measured from the trough of the a-wave to the peak of the *b*-wave.

### 4.6. Histology

Eyes were enucleated from euthanized mice, followed by cautery on the limbus to create a mark at the 12 o’clock position. H&E histology was carried out as previously described [45]. In summary, after the fixation and paraffin section, hematoxylin and eosin were used to stain the paraffin sections, which were then examined by light microscopy. Imaging was taken 200 μm from the optic nerve head.

### 4.7. Immunoblotting

Western blot analysis was performed as described previously [45,46,47]. In summary, retinas from 8-week-old mice were homogenized in RIPA lysis and extraction buffer (Prod #89900 Thermo Scientific, Waltham, MA, USA) containing protease inhibitor cocktail (catalog P8340-1 mL; Sigma, St. Louis, MO, USA), and separated by 4–15% SDS polyacrylamide gel (Prod#4561083) electrophoresis as previously described. After the separation of the proteins, the proteins on SDS-PAGE were transferred to nitrocellulose membranes (Bio-Rad, Dreieich, Germany). The nitrocellulose membranes were subsequently blocked in 4% skim milk (902887 MP Biomedicals, LLC, Solon, OH, USA) dissolved in phosphate buffered saline and 0.1% Tween 20 (PBST) for 1 h. After blocking and washing, the nitrocellulose membranes were proceeded to overnight incubation at 4 °C with primary antibody (anti-Cnga3, 1:2000, orb625890, Biorbyt Ltd., Cambridge, UK; anti-Arr3, 1:500, PA5-49513, ThermoFisher Scientific, Waltham, MA, USA; anti-Cngb3, 1:1000, orb500408, Biorbyt Ltd., Cambridge, UK; anti-β-actin 1:2500, 8H10D10, Cell Signaling Technology, Danvers, MA, USA, or anti-GAPDH 1:2500, D16H11, Cell Signaling Technology), respectively. After rinsing in 0.1% PBST, the nitrocellulose membranes were incubated with mouse monoclonal anti-rabbit IgG-HRP secondary antibody (1:3000; ab99697; Abcam, Waltham, MA, USA) or rabbit anti-mouse IgG-HRP secondary antibody (1:3000, ab6725; Abcam) for 1 h at room temperature. Chemiluminescent substrate (Immobilon Western, Millipore Corporation EMD Millipore, Burlington, MA, USA) was used to detect the binding of the primary antibodies to their cognate antigens. Signals were visualized and captured using the iBright Imaging System (FL1500; Invitrogen).

### 4.8. Quantitative Real-Time RT-PCR (qRT-PCR)

Total RNA was extracted from freshly dissected mouse retinas using the Invitrogen TRIzol reagent (ThermoFisher Scientific, 15596026). The RNA was reverse—transcribed to cDNA by using SuperScript III First-Strand Synthesis (Invitrogen, 18080-400). qPCR amplification was run in a final concentration of 20 μL by using the SsoAdvanced Universal SYBR Green Supermix (Bio-Rad, 1725271) on the BIORAD CFX Connect Real-Time PCR system. The primer pairs for each gene in this study are listed in Table 3. Reactions were performed with the following thermocycling settings: An initial denaturation step of 95 °C for 3 min, followed by 38 cycles of 95 °C for 8 s, an annealing temperature of 60 °C, and extension at 72 °C for 20 s. The expression of target genes was normalized to *Rhopdosin.*

### 4.9. Statistical Analysis

All experimental data were assessed by an operator blinded to the genetic condition. Statistical significance was determined by independent student *t*-test and one-way ANOVA for multiple comparison. *p* < 0.05 was considered significant. All analyses were performed using SPSS (IBM SPSS Statistics for Windows, Version 21.0, IBM Corp., Armonk, NY, USA) and GraphPad Prism 9.

## 5. Conclusions

In summary, although CNGA3—together with CNGB3—contributes the major parts of the CNG protein channel, where many achromatopsia patients carry mutations in both genes, merely having a small amount of normal CNGA3 protein is sufficient to form “functional” CNG channels. Theoretically, the last exon (sixth exon) of the *Cnga3* gene contributes most of the CNGA3 protein that forms the CNG channel; thus, placing a “*floxed-STOP*” cassette in front of the sixth exon is the ideal choice to create a reversible mouse model of achromatopsia. However, the splicing score of the sixth exon in *Cnga3* is too robust to allow the termination of transcription by the inserted “*floxed-STOP*” cassette, and the alternative strategy of mutating the *Cnga3* locus to disrupt functional tetrameric CNG channels is a challenging task. Future studies may want to explore more powerful STOP cassettes in order to target the *Cnga3* gene and create a reversible mouse model of achromatopsia that more accurately recapitulates the human phenotype.

From our study, we found the cone ERG response corresponds to the normal cone phototransduction, which requires “functional” CNG channels from normal membrane proteins, rather than residual cone cells or “levels” of cone proteins. Our data suggests that only 20% of CNGA3 protein levels are needed in order to achieve physiological demand, and is thus sufficient for normal cone phototransduction. This was confirmed by ERG in our *Cnga3^floxed-STOP^* mice.

Throughout the study, we learned several valuable lessons on the limitations that may arise when conducting such research. First, while generating genetically modulated mouse models using *Easi*-CRISPR is more time- and cost-efficient compared to targeted ES cells, this technique is limited by the size of the knock-in gene fragments (less than 1kb). Second, CRISPR/Cas9 may potentially result in large deletions, which could extend over many kilobases. Further refinement of CRISPR/Cas9 may improve the safety of any direct in vivo applications, prior to translating this technology into a gene therapy product. Lastly, given that no standardized ERG protocol exists for conducting mouse ERGs, it becomes quite difficult to directly compare ERG results across separate studies conducted by varying laboratories.

Although we were not successful in creating reversible mouse models of achromatopsia, our findings remain encouraging for future clinical trials. In particular, our data may imply that in future clinical trials, restoration of 20% functional CNGA3 protein may form functional CNG channels and rescue the cone response in *CNGA3*-associated achromatopsia. In addition, there is a need for standardized ERG protocols for characterizing electrophysiologic function in mice in the future.

## Figures and Tables

**Figure 1 ijms-22-08069-f001:**
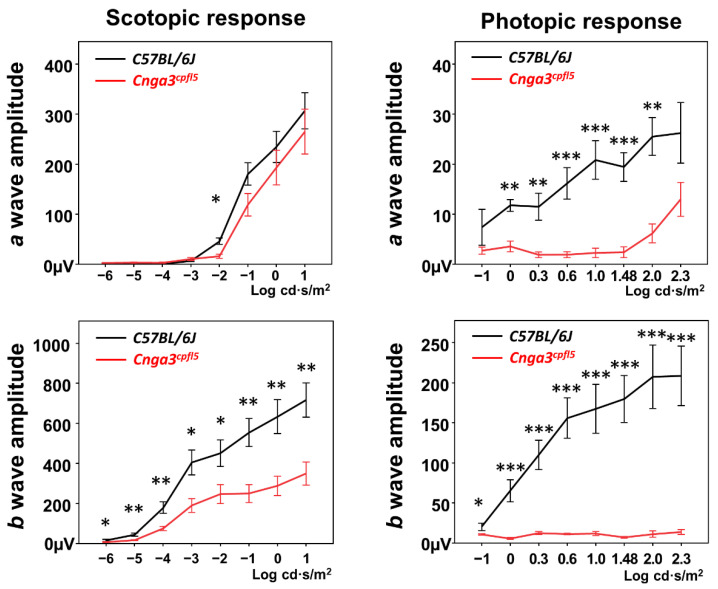
Electroretinography (ERG) of *Cnga3^cpfl5^* mice at one month old. (**Left**) Scotopic serial intensity ERG tracings showed statistically significant decreased *b*-wave amplitudes in *Cnga3^cpfl5^* mice compared to age-matched C57BL/6J mice. (**Right**) Photopic serial intensity ERG in *Cnga3^cpfl5^* mice showed nearly extinguished *b*-wave amplitudes and statistically significantly decreased *b*-wave amplitudes compared to the *b*-wave amplitudes in age-matched C57BL/6J mice. (n*_Cnga3cpfl5_* = 5, n*_C57BL/6J_* = 3). * *p* < 0.05, ** *p* < 0.01, *** *p* < 0.001.

**Figure 3 ijms-22-08069-f003:**
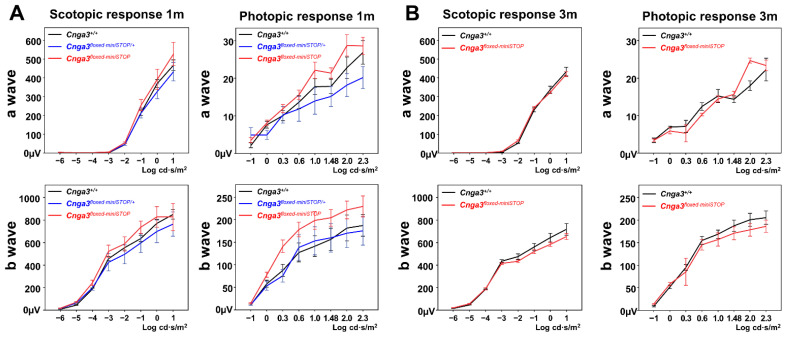
Electroretinography (ERG) of *Cnga3^floxed-miniSTOP^* mice. (**A**) Scotopic and photopic serial intensity ERG tracings of *Cnga3^floxed-miniSTOP^* mice showed no decrease in *b-*wave amplitudes compared to the *Cnga3^+/+^ and Cnga3^floxed-miniSTOP/+^* mice from the same litter at 1-month old (n*_Cnga3+/+_* = 3, n*_Cnga3floxed-miniSTOP/+_* = 2, n*_Cnga3floxed-miniSTOP_* = 5). (**B**) Scotopic and photopic serial intensity ERG of *Cnga3^floxed-miniSTOP^* mice showed no decrease in *b*-wave amplitudes compared to the *Cnga3^+/+^* mice from the same litter at 3 months old (n*_Cnga3+/+_* = 3, n*_Cnga3floxed-miniSTOP_* = 3).

**Figure 4 ijms-22-08069-f004:**
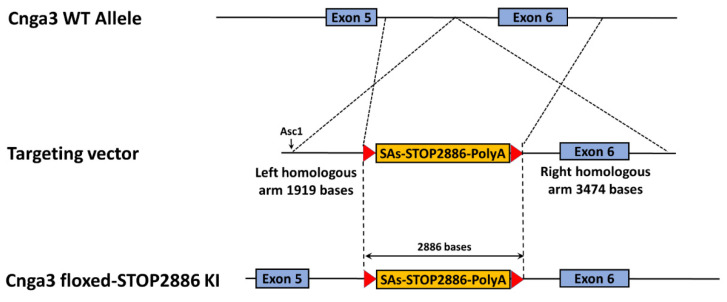
Generation of *Cnga3^floxed-STOP^* mice using embryonic stem (ES) cell gene targeting. We inserted a “*floxed*-*STOP*” (2886 base pairs) cassette into the intron between exon 5 and exon 6 of the *Cnga3* gene using ES cell gene targeting.

**Figure 5 ijms-22-08069-f005:**
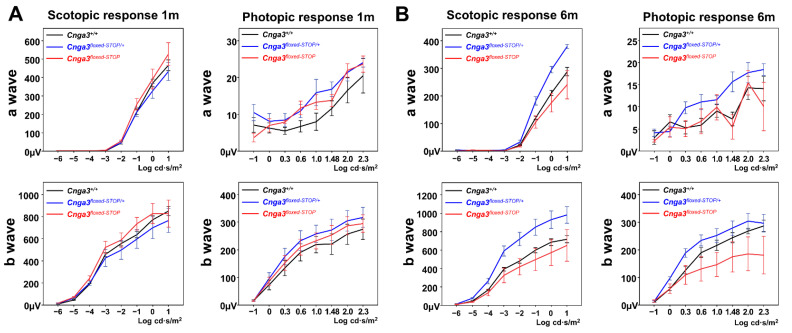
Electroretinography (ERG) of *Cnga3^floxed-STOP^* mice. (**A**) Scotopic and photopic serial intensity ERG tracings of *Cnga3^floxed-STOP^* mice showed no decrease in *b*-wave amplitudes compared to the *Cnga3^+/+^* and *Cnga3^floxed-STOP/+^* mice from the same litter at 1-month old. (n*_Cnga3+/+_* = 5, n*_Cnga3floxed-STOP/+_* = 4, n*_Cnga3floxed-STOP_* = 6) (**B**) Although the photopic b-wave amplitudes are smaller in *Cnga3^floxed-STOP^* mice at 6 months old, the difference is not statistically significant. (n*_Cnga3+/+_* = 3, n*_Cnga3floxed-STOP/+_* = 3, n*_Cnga3floxed-STOP_* = 5).

**Figure 6 ijms-22-08069-f006:**
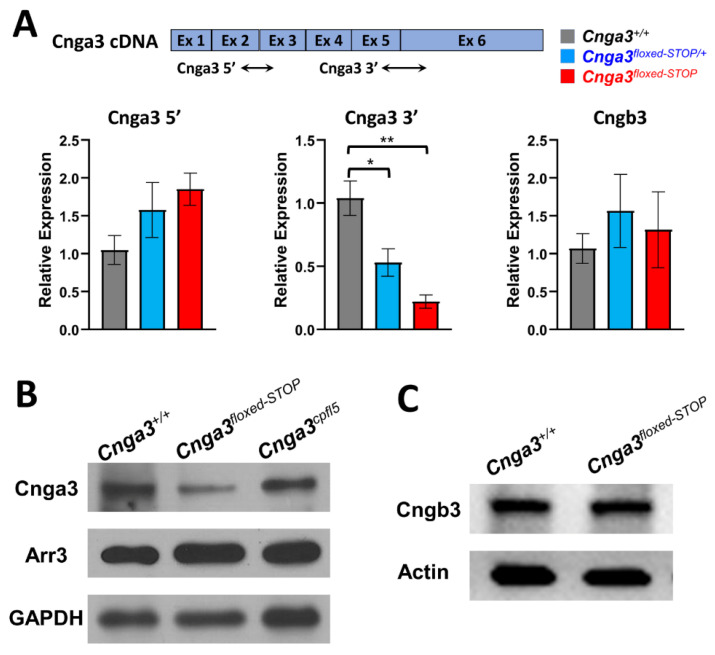
Quantitative Real-Time RT-PCR and Western blotting of *Cnga3^floxed-STOP^* mice. (**A**) Two sets of primer pairs were designed to span intron 2 and intron 5, respectively, in order to identify the presence of changes in expression of *Cnga3* transcript in *Cnga3^floxed-STOP/+^* (blue) and *Cnga3^floxed-STOP^* (red) mice. *Cnga3^+/+^* littermates (grey) were included as the wild-type (WT) controls. Despite an increasing trend, the level of *Cnga3* amplicon upstream of the *floxed*-*STOP* cassette lacks statistical significance, whereas the level of *Cnga3* amplicon downstream of the *floxed*-*STOP* cassette was significantly reduced in *Cnga3^floxed-STOP/+^* and *Cnga3^floxed-STOP^* mice. The expression of target genes was normalized to mouse *Rhopdosin.* Results = mean ± S.E.M (n*_Cnga3+/+_* = 5, n*_Cnga3 floxed-STOP/+_* = 4, n*_Cnga3 floxed-STOP_* = 3; * *p* = 0.0294; ** *p* = 0.0032). (**B**) There was reduced CNGA3 protein expression in *Cnga3^floxed-STOP^* mice compared to littermate controls, while there was no significant decrease in CNGA3 protein expression in *Cnga3^cpfl5^* mice. (**C**) There was no obvious change of CNGB3 protein between *Cnga3^floxed-STOP^* and littermate controls.

**Figure 7 ijms-22-08069-f007:**
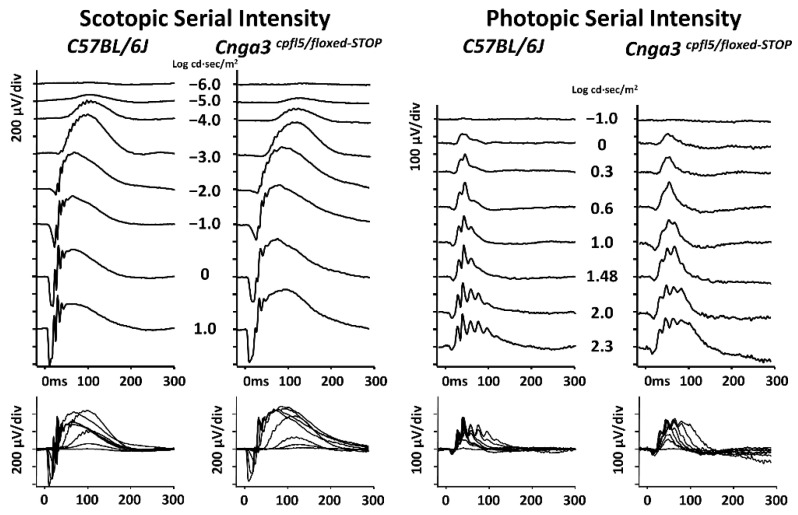
Electroretinography (ERG) of compound heterozygous *Cnga3^cpfl5/floxed-STOP^* mice. The scotopic and photopic serial intensity ERG responses from these compound heterozygous mice showed relatively normal ERG responses as compared to C57BL/6J mice at the same age (postnatal day 31, P31).

**Figure 2 ijms-22-08069-f002:**
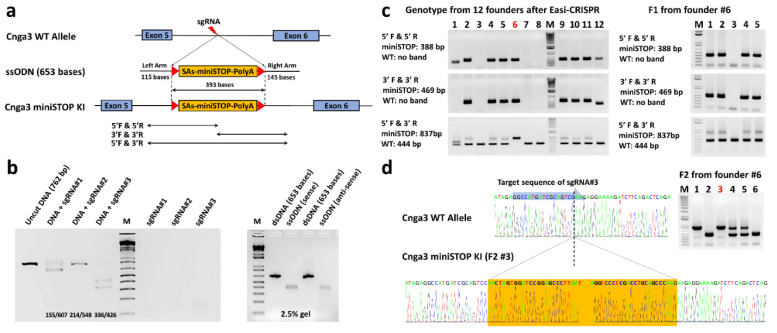
Generation of *Cnga3^floxed-miniSTOP^* mice using *Easi*-CRISPR. (**a**) A *floxed-miniSTOP* cassette (393 bps) was knocked into intron 5 of the *Cnga3* gene using *Easi*-CRISPR. The ssODN (653 bases) includes 393 bases of the *floxed-miniSTOP* cassette and homologous sequences at both ends (115 bases and 145 bases). Primers were designed to screen the 5′ and 3′ junction of the knock-in floxed-miniSTOP cassette. (**b**, **left**) In vitro digestion (IVD) assay showed the best cutting efficiency with sgRNA#3. (**b**, **right**) Gel image of dsDNA and ssODN, which include 393 bases of the *floxed-miniSTOP* cassette (393 bps) and homologous sequences at both ends corresponding to the cutting site of sgRNA#3. (**c**, **left**) Polymerase chain reaction (PCR) results of twelve founders using primers to screen the 5′ and 3′ junction of the miniSTOP and identified 7 mice (#2, 4, 5, 6, 9, 10, 11) of the 12 founders carry the knock-in floxed-miniSTOP cassettes. PCR was performed using primers 5′F and 3′R to screen for the wild-type (WT) and knock-in floxed-miniSTOP cassette allele in these 12 founders and found that only the knock-in floxed-miniSTOP cassettes allele could be amplified from founder #6. (**c**, **right**) PCR screening of the five F1 mice revealed four F1 mice carried the knock-in floxed-miniSTOP cassette allele. The F1 animals (#1 and #5) were intercrossed to produce homozygous F2 offspring to confirm penetrance. (**d**, **right**) PCR screening using primers 5′F and 3′R on six F2 mice showed two WT, two heterozygous, and two homozygous of *Cnga3^floxed-miniSTOP^* mice. (**d**, **bottom**) Direct sequence from #3 F2 mice showed successfully knock-in floxed-miniSTOP cassette in the intron 5 of the *Cnga3* gene.

**Table 1 ijms-22-08069-t001:** Sequence of sgRNAs.

sgRNA	Strand	Sequence (5′-3′)	PAM	Off-Target Score	On-Target Score
*sgRNA#1*	Forward	gcagtcacactagtatccac	agg	42.8263215	66.09532194
*sgRNA#2*	Forward	cttgtctcaggcgagcctca	ggg	42.2064369	55.07114307
*sgRNA#3*	Forward	ggccatgatcgcagtccaag	agg	45.198453	63.29196792

**Table 3 ijms-22-08069-t003:** Primers used in the transcriptome analysis.

Gene	Direction/Exon	Nucleotide Sequence (5′-3′)
*Cnga3*	Forward/5	ATGAGCTACAATCAGAGCACC
	Reverse/6	TTTCCACAGCCTCTTGGTATC
*Cnga3*	Forward/2	CCCCGACCCAACTTTCAATA
	Reverse/3	TTCATCGTGTAAGTGCCTGG
*Cngb3*	Forward/2	GCAGGACACCAATCACATTTG
	Reverse/3	CCTAGTTTTCTCCATCTCTGCC
*Rhodopsin*	Forward/2	GGGAGAATCACGCTATCATGG
	Reverse/3	GTCAATCCCGCATGAACATTG
*Beta-Actin*	Forward/3	ACCTTCTACAATGAGCTGCG
	Reverse/4	CTGGATGGCTACGTACATGG

**Table 4 ijms-22-08069-t004:** Predicted scores of splicing acceptors using two different algorithms.

***Cnga3***			
**Exon**	**Position**	**Score 1 ***	**Score 2 #**
1	1–278	0	0
2	12949–13085	0.95	0
3	25573–25752	0.65	0.97
4	34697–34804	0.96	0.92
5	38733–38839	0.95	0.86
6	41437–44124	0.97	0.99
***Cnga3 floxed-STOP***			
**Exon**	**Position**	**Score 1 ***	**Score 2 #**
1	1–278	0	0
2	12949–13085	0.95	0
3	25573–25752	0.65	0.97
4	34697–34804	0.96	0.92
5	38733–38839	0.95	0.86
floxed-STOP cassette	41121–44006	0.97	1
6	44323–47010	0.97	0.99

*: http://www.cbs.dtu.dk/services/NetGene2/; #: https://www.fruitfly.org/seq_tools/splice.html; (both accessed on 26 June 2021).

**Table 2 ijms-22-08069-t002:** Primers for genotype knock-in mice.

Gene	Direction	Nucleotide Sequence (5′-3′)	Size
*Floxed-miniSTOP*	5′ Forward	GTATGAATGGCCTCTGCAGCAGTCA	388
	5′ Reverse	ACTGGAAAGACCGCGAAGAGTTTGT	
*Floxed-miniSTOP*	3′ Forward	CTCTTCGCGGTCTTTCCAGTTAACT	469
	3′ Reverse	GAAGCCGTATTGTGTTGTGTAGGGA	
*Floxed-STOP*	5′ Forward	CACTGTTTCCCTGAACACTACT	285
	5′ Reverse	AGACAGGGTCTTGTATAGGCTACATTC	
*Floxed-STOP*	3′ Forward	GGGAGGATTGGGAAGACAATAG	355
	3′ Reverse	TTTCTCTTCTGGGAATGGAACC	

## Data Availability

Data are contained within the article or Appendix A.

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
