# Peer review of "Mouse Models of Achromatopsia in Addressing Temporal “Point of No Return” in Gene-Therapy"

_ijms, 2021, doi:10.3390/ijms22158069_

Round 1

Reviewer 1 Report

The authors of the present study describe the generation of two different mouse models of achromatopsia, i.e., CNGA3 mutants for further analysis of the treatment time window for gene therapeutic approaches. The background and purpose of the study are well described and the results are clearly and comprehensible presented supporting the conclusion. However, mainly the title and the abstract are misleading, since finally the generation of reversible mouse models did not work and the generated mice elicit normal ERG function in most cases. The results of the study are nevertheless of interest for the reader and should be shared; however, the discrepancy in the title and abstract to the main article should be resolved.

In general, wording and English of the manuscript are very good.

The introduction is well structured showing a clear golden thread; it is detailed enough without being too long.

The results are very well structred and explained and thus, it is easy to understand and follow the story. The figures are of good quality and sufficiently described.

The methods are well-structured as well giving sufficient details for potential reproduction.

The discussion analyzes nicely why the authors concluded (including the comparison with other models and literature) that not the number of (dead/lost) cone receptors, but the number of functional channels seems to be crucial for a normal b-wave and why they suggest that only 20% of functional channels (for which sufficient CNGA3 protein is necessary) are sufficient for normal vision.

Major comment

However, as already mentioned in my summary, though the whole article is well written, I expected to read something different.

Please modify the title by deleting “reversible”. Also the dimension of time “promises more than will be delivered later”, why I suggest to change this, too.

Then, in the abstract it is written “we generated two recoverable mouse models…”, which is simply not the case. Please modify the abstract accordingly and emphasize more on the surprising result that only 20% of functional CNGA3 protein are sufficient to form functional channels to rescue cone phototransduction.

In the main article this “misleading wording” is not as strong as in the title and the abstract, but also present except in the conclusion. Thus, please review the whole article regarding this issue.  

Reviewer 2 Report

Aim of the work was to produce a reversible genetic model of achromatopsia, by inserting an inducible stop codon within one of the genes forming cone membrane channels necessary for cone functioning. However, the method used (CRISPR-CAS9) failed to produce homogeneous knock-in models, so that some 20% of correct protein could still be produced, enough to guarantee a correct functioning of cones in transgenic animals. Such negative results have nonetheless an interesting meaning, in that they help to establish a minimum amount of channel protein required for cone functioning, and from a methodological perspective they can help to understand the pitfalls of the CRISPR-CAS9 methodology, and improve further experimental designs. Therefore, I believe that the study deserves publication and to be spread among the scientific community.

Reviewer 3 Report

Dear Authors,

presented paper shows for the first time orginal  mouse model of achromatopsia with aplication of  Easi-CRISPR technology. 

The study is well designed, methodology is apropriatly designed and described with clear presentation of results functional results.

Alghough in text of manuscript some limitation of the methodology and achieved results are suggested, but they are scattered in the manuscript. Therefore I suggest additional paragraph or summary with limitation of the study: it is very usefull for readers to have a paragraph highligting the streights and the drawbacks of the following study.

I recommend to reconsider it after minor revision for the publication.

Round 2

Reviewer 1 Report

The authors responded well to the comments of the first review round and I have no more comments on the revised version of the manuscript.